# Coaxial 3D Bioprinting Process Research and Performance Tests on Vascular Scaffolds

**DOI:** 10.3390/mi15040463

**Published:** 2024-03-29

**Authors:** Jiarun Sun, Youping Gong, Manli Xu, Huipeng Chen, Huifeng Shao, Rougang Zhou

**Affiliations:** 1School of Mechanical Engineering, Hangzhou Dianzi University, Hangzhou 310018, China; 231010006@hdu.edu.cn (J.S.); gyp@hdu.edu.cn (Y.G.); hpchen@hdu.edu.cn (H.C.); shaohf@hdu.edu.cn (H.S.); 2State Key Laboratory of Fluid Power and Mechatronic Systems, School of Mechanical Engineering, Zhejiang University, Hangzhou 310027, China; 3The 2nd Affiliated Hospital of Zhejiang Chinese Medical University, Hangzhou 310005, China; 4Jiangsu Key Laboratory of 3D Printing Equipment and Manufacturing, Nanjing Normal University, Nanjing 210042, China; 5Mstar Technologies, Inc., Room 406, Building 19, Hangzhou Future Science and Technology City (Haichuang Park), No. 998, Wenyi West Road, Yuhang District, Hangzhou 311121, China

**Keywords:** 3D bioprinting, coaxial jet, vascular network, finite element analysis, biological scaffold

## Abstract

Three-dimensionally printed vascularized tissue, which is suitable for treating human cardiovascular diseases, should possess excellent biocompatibility, mechanical performance, and the structure of complex vascular networks. In this paper, we propose a method for fabricating vascularized tissue based on coaxial 3D bioprinting technology combined with the mold method. Sodium alginate (SA) solution was chosen as the bioink material, while the cross-linking agent was a calcium chloride (CaCl_2_) solution. To obtain the optimal parameters for the fabrication of vascular scaffolds, we first formulated theoretical models of a coaxial jet and a vascular network. Subsequently, we conducted a simulation analysis to obtain preliminary process parameters. Based on the aforementioned research, experiments of vascular scaffold fabrication based on the coaxial jet model and experiments of vascular network fabrication were carried out. Finally, we optimized various parameters, such as the flow rate of internal and external solutions, bioink concentration, and cross-linking agent concentration. The performance tests showed that the fabricated vascular scaffolds had levels of satisfactory degradability, water absorption, and mechanical properties that meet the requirements for practical applications. Cellular experiments with stained samples demonstrated satisfactory proliferation of human umbilical vein endothelial cells (HUVECs) within the vascular scaffold over a seven-day period, observed under a fluorescent inverted microscope. The cells showed good biocompatibility with the vascular scaffold. The above results indicate that the fabricated vascular structure initially meet the requirements of vascular scaffolds.

## 1. Introduction

The body’s organs are subject to wear and deterioration due to aging and disease. The cardiovascular system is susceptible to specific conditions, such as cerebrovascular disease or peripheral arterial disease, which reduce blood flow to various areas of the body [1,2]. In many countries, cardiovascular disease is the leading cause of death, far outnumbering cancer [3]. Data estimate that by around 2030, the annual death rate from cardiovascular disease will rise to 23.6 million worldwide [4,5]. Therefore, the treatment of vascular diseases has become a crucial aspect of human health nowadays.

Currently, the primary treatment for cardiovascular diseases is vascular replacement, which includes autologous blood vessels, allogeneic blood vessels, and artificial blood vessels. Autologous blood vessels are obtained from the patient’s own blood vessels, usually located near the lesion. This is the most common method and source due to the good compatibility and ability to fuse well with the patient’s own organs and tissues. However, there are some significant drawbacks. Firstly, the entire operation time is very long and the wound area is large, which can potentially cause secondary injuries to the patient. Additionally, the number of available blood vessels in the patient’s body is limited, making it impossible to obtain them multiple times, which can result in the failure of the operation. Allogeneic vascular grafts are highly susceptible to complications and rejection due to incompatibility, which can cause further harm to the human body. Artificial blood vessels are mainly blood vessels fabricated from traditional materials, most of which are polyester or some polymer materials lacking biological activity. Patients transplanted with these blood vessels need to take long-term medication to prevent blood coagulation and the formation of blood clots and are also prone to endothelial hyperplasia at the junction of small-diameter blood vessel grafts, which can lead to vascular occlusion [6,7,8]. Conventional methods for artificial vascular construction include vacuum freeze-drying [9,10], thermotropic phase separation [10,11,12,13,14,15], and electrostatic spinning [16,17,18].

Compared to non-degradable artificial blood vessels, the utilization of biomaterials to fabricate degradable tissue engineering scaffolds through 3D bioprinting technology represents a more desirable approach. It is widely used in the fabrication of engineered tissue constructs. With the advancement of 3D bioprinting technology, the utilization of 3D printing in the fabrication of tissue-engineered blood vessels has become increasingly prevalent. Most methods for creating vascular channels are based on sacrificial printing, which typically involves four steps: printing the sacrificial biostructure, embedding the printed sacrificial biostructure into a hydrogel matrix through a molding process, unraveling the sacrificial biostructure to form a hollow vascular channel, and implanting endothelial cells on the surface of the hollow vascular channel by perfusing an endothelial cell suspension. Raymond et al. [19] developed a coaxial nozzle-assisted embedded 3D printing method to directly print vascular structures within a matrix material. However, endothelial cells later perfused are difficult to successfully seed in the complex vascular structure. He Yong’s team at Zhejiang University proposed a method for fabricating vascularized tissue on a coaxial jet [20]. The method involved using a bioink prepared with sodium alginate and divalent cations, such as calcium chloride, for cross-linking to create a tube channel. The coaxial nozzle had an alginate-based bioink in the outer runner and a cross-linking agent in the inner runner. The 3D printing equipment was then used to deposit the continuous hydrogel tubes, and implanted into endothelial cells by perfusion. However, as the number of printed layers increased, the force of gravity could cause the lower layer of the tube to collapse.

Bioink is a revolutionary new term recently introduced in the development of artificial 3D tissues or organs. The application of bioink, which has achieved significant progress in medical applications, has enabled the 3D bioprinting technology to process live cells, biomaterials, and other configurable materials. This technology allows for precise control over the type and quantity of cells, enabling the creation of complex and customized tissue or organs through the printing and fabrication process. The primary methods employed in 3D printing for blood vessels include microdroplet jetting [21], the coaxial jetting method, and the mold method [22,23]. Wu et al. [24] wrapped human umbilical vein smooth muscle cells (HUVSMCs) in sodium alginate and successfully printed them in the form of blood vessel ducts using a coaxial nozzle deposition system. Unlike traditional tissue engineering methods, in which cells are inoculated after scaffold fabrication, bioprinting technology enables cells to be inoculated during the fabrication process by depositing them within the scaffold matrix. This is a 3D layer-by-layer bio-addition process in which a computer program-controlled bioprinter is used to print a high density of living cells [25,26].

In recent years, there has been significant research interest in developing methods for fabricating vascularized tissue. The microdroplet jetting method allows for the encapsulation of cells within a matrix material, theoretically enabling the printing of complex vascularized tissue. However, this method demands stringent fabrication environments and technical requirements, coupled with intricate fabrication processes that are challenging to operate. The coaxial 3D bioprinting method has the potential to print various tubular structures, making it particularly suitable for creating blood vessels. The resulting vascular scaffolds possess the necessary structure and strength to meet requirements. However, this method has the limitation of not being able to fabricate complex vascular networks. At the same time, the mold method allows for the fabrication of more complex structures, but it also has inherent disadvantages. Firstly, the selection and fabrication of the mold are crucial. If materials are chosen improperly, it may result in challenges during demolding and potential damage to the hydrogel tubes. Secondly, different product shapes require different molds, and the process of manufacturing molds is also expensive.

In this paper, we have primarily conducted theoretical and simulation analyses on the fluid behavior of bioink during the coaxial jetting process, as well as the mechanical behavior of vascular stents during blood transportation. These studies aimed to further determine the process parameters of the coaxial printing process and validate the performance of the vascular scaffolds. Subsequently, we fabricated multiple sets of vascular scaffolds under different process parameters to optimize them. Additionally, we successfully attempted the fabrication of vascular network structures by combining them with the mold method, which effectively circumvented the disadvantages of the two separate fabrication techniques. Therefore, combining the coaxial jetting method with the mold method to fabricate tissue structures with vascular networks introduces new research directions and ideas for related fields.

## 2. Materials and Methods

### 2.1. Selection and Preparation of Materials

The attractive properties of alginate as a biomaterial include non-immunogenicity, rapid cross-linking, low toxicity, and good biodegradability and biocompatibility [27]. Sodium alginate is a low-cost, easy-to-handle substance that is highly susceptible to cross-linking reactions with Ca^2+^ ions, resulting in the formation of calcium alginate. Therefore, in this experiment, sodium alginate (Shanghai Aladdin Biochemical Technology Co., Ltd., Shanghai, China) was used as a bioink material, and calcium chloride (Shanghai Aladdin Biochemical Technology Co., Ltd., Shanghai, China) was used as a cross-linking agent. The configuration procedure is as follows:Weigh the solid reagent particles using an electronic balance (LICHEN, Changsha, China) before use;Pour the sodium alginate particles into batches and dissolve them in deionized water. The constant temperature magnetic stirrer (Shanghai Sile Instrument Co., Ltd., Shanghai, China) was used to stir the mixture, ensuring complete dissolution;Dissolve the solid calcium chloride particles in deionized water by stirring with a glass rod for 5 min until fully dissolved;Sterilize the prepared sodium alginate solution first, and then inject the HUVEC (Shenggong Biotechnology (Shanghai) Co., Ltd., Shanghai, China) suspension into the prepared sodium alginate solution to complete the preparation of the bioink. Finally, keep it at room temperature before use.

### 2.2. Modeling the Fluid Behavior of Bioink

The power law fluid suspension model, as a simple model describing the use of rigid particles as encapsulated cells, allows for the study of the behavior of biological inks at the threshold of instability. This theoretical model can be adjusted to the onset of instability by varying the concentration of rigid particles:(1)$$\mu_{AV}=\frac{\tau }{\gamma }=K\gamma^{n-1}$$
(2)$$\tau =K\gamma^{n}$$where $\mu_{AV}$ denotes the apparent viscosity, τ is the shear stress, γ denotes the shear rate, *K* denotes the viscosity coefficient, and *n* denotes the power flow exponent. The fluid is a Newtonian fluid when *n* = 1, a pseudoplastic fluid when *n* < 1, and an expansive fluid when *n* > 1. Sodium alginate solution viscosity decreases with the increase in shear force, so sodium alginate solution is pseudoplastic fluid, which is analyzed by the power law fluid model with the value of the power law exponent *n* < 1.

### 2.3. Theoretical Model Analysis and Numerical Simulation of Coaxial Jet Process

The bioink enters the outer runner through the outer nozzle inlet and the cross-linking agent enters the inner runner through the inner needle inlet. Both flow through the lumen and out at the nozzle outlet, where the cross-linking reaction occurs. As shown in Figure 1, the fluid flow inside the nozzle is divided into two parts: internal and external.

This study analyzes three aspects: the theoretical model of fluid material flow inside the nozzle prior to exiting, the process of two materials flowing out of a coaxial nozzle and cross-linking with the fluid, and the analysis of the theoretical model of fluid flow after cross-linking. The fluid dynamics are governed by the laws of conservation of mass and conservation of linear momentum. To simplify our simulations, we have made the following assumptions regarding the physical properties of the fluid flow:At constant density, the material is considered incompressible: conservation of mass translates into conservation of volume;The fluid is assumed to be peristaltic, and inertial effects are negligible compared to viscous forces;Bioink materials are modeled using non-Newtonian mechanical behavior, where the viscosity of the deposited fibers can be defined in terms of a power law exponent;Extrusion-based bioprinting is temperature-independent, and fiber deposition is considered isothermal.

#### 2.3.1. Theoretical Analysis of Coaxial Nozzle Fluid Model

The theoretical analysis of the hydrodynamic model of calcium chloride solution in the inner nozzle is shown in Figure 1, a micro-volume unit within the inner runner is selected for mechanical analysis, which is subjected to a combination of viscous force, pneumatic pressure, and gravity. Equation (3) is obtained from the Newton’s second law equation as follows:(3)$$\begin{array}{c} 2\pi \left[rPdr-r\left(P+\frac{\partial P}{\partial x}dx\right)dr+r\tau_{1}dx-\left(r+dr\right)\left(\tau_{1}+\frac{\partial \tau_{1}}{\partial r}dr\right)dx+r\rho_{1}gdxdr\right]\\ =2\pi r\rho_{1}adxdr \end{array}$$
(4)$$a=\frac{du_{1}}{dt}$$

In the above Equations (3) and (4), *P* is the pressure, $\tau_{1}$ is the viscous shear stress, $\rho_{1}$ is the density of calcium chloride solution, *r* is the horizontal coordinate of the micro-volume, *x* is the vertical coordinate of the micro-volume, $\mu_{1}$ is the viscosity of calcium chloride solution, *g* is the acceleration of gravity, $u_{1}$ is the velocity of flow of the micro-volume in the inner runner, and *a* is the micro-volume acceleration. Equations (5) and (6) can be obtained as follows:(5)$$\rho_{1}g-\frac{\partial \left(\tau_{1}r\right)}{r\partial r}-\frac{\partial P}{\partial x}=\rho_{1}a$$
(6)$$\frac{\partial P}{\partial x}=-\frac{\Delta P_{1}}{L_{1}}$$

In practical engineering problems, the pressure drop (Δ*P*) per unit length of flow distance (*L*) is commonly used in place of the pressure gradient. In this paper, it is assumed that the fluid outflow process maintains a uniform velocity, so that the acceleration is zero. Since calcium chloride solution is a Newtonian fluid, Equation (7) can be obtained as follows:(7)$$\tau_{1}=-\mu_{1}\frac{du_{1}}{dr}$$

From Equations (5)–(7), Equation (8) can be obtained as follows:(8)$$\frac{\partial }{\partial r}\left(r\frac{du_{1}}{dr}\right)=-\frac{\left(\Delta P_{1}+\rho_{1}gL_{1}\right)r}{\mu_{1}L_{1}}$$

The fluid satisfies the no-slip condition at the wall during flow, *r* = *R* (*R* is the inner radius of the inner nozzle) and $u_{R}$ = 0, and Equation (9) can be obtained by integrating Equation (8) as follows:(9)$$u_{1}=\frac{\left(\Delta P_{1}+\rho_{1}gL_{1}\right)\left(R^{2}-r^{2}\right)}{4\mu_{1}L_{1}}$$

It is seen that the velocity is maximized when the fluid passes through the axis (*r* = 0):(10)$$u_{1max}=\frac{\left(\Delta P_{1}+\rho_{1}gL_{1}\right)R^{2}}{4\mu_{1}L_{1}}$$

And from Equation (9), it can be seen that the velocity of the fluid in the axial section of the inner runner is a quadratic curve, showing a parabolic distribution. Substituting Equation (9) back to Equation (7), Equation (11) can be obtained as follows:(11)$$\tau_{1}=\frac{\left(\Delta P_{1}+\rho_{1}gL_{1}\right)r}{2L_{1}}$$

From Equation (11), it can be seen that the shear stress received by the micro-volume is zero at the axis (*r* = 0), and since $\left(\Delta P_{1}+\rho_{1}gL_{1}\right)/2L_{1}$ is a constant, the viscous shear stress on any cross-section of the circular tube satisfies the linear distribution relationship on the *r*-axis, and the fluid receives the maximum viscous shear stress near the wall.

The above theoretical derivation is also applicable to the hydrodynamic modeling of sodium alginate solutions in outflow channels, as shown in Figure 1. However, since sodium alginate solution is a non-Newtonian fluid, then the shear stress $\tau_{2}$ expressed as Equations (8) and (12) is rewritten as Equation (13):(12)$$\tau_{2}=-\mu_{2}\frac{du_{2}}{dr}=-K{\left(\frac{du_{2}}{dr}\right)}^{n}$$
(13)$$\frac{\partial }{\partial r}\left[{r\left(\frac{\partial u_{2}}{\partial r}\right)}^{n}\right]=-\frac{\left(\Delta P_{2}+\rho_{2}gL_{2}\right)r}{KL_{2}}$$where $\tau_{2}$ is the viscous shear stress, $\rho_{2}$ is the density of sodium alginate solution, *r* is the horizontal coordinate of the micro-volume,$\;\mu_{2}$ is the apparent viscosity of sodium alginate solution, K is the viscosity coefficient of sodium alginate solution, *g* is the acceleration of gravity, $u_{2}$ is the velocity of flow of the micro-volume in the outer runner, $\Delta P_{2}$ is the pressure drop in the outer runner, $L_{2}$ is the length of flow distance in the outer runner, and *n* is the power flow exponent.

$R_{1}$ is the inner diameter of the outer nozzle, $R_{2}$ is the outer diameter of the inner nozzle, and the boundary conditions are $r_{max}=R_{1}$, $r_{min}=R_{2}$. The flow model satisfies the condition of no slip at the wall, so $r=R_{2}$, $u_{R_{2}}=0$; $r=R_{1}$, $u_{R_{1}}=0$. The result is obtained by integrating Equation (13):(14)$$u_{2}=\frac{n}{n+1}{\left[\frac{\left(\Delta P_{2}+\rho_{2}gL_{2}\right)}{2KL_{2}}\right]}^{\frac{1}{n}}\left[{R_{1}}^{\frac{1}{n}+1}-r^{\frac{1}{n}+1}+\frac{\ln\frac{r}{R_{1}}\left({R_{1}}^{\frac{1}{n}+1}-{R_{2}}^{\frac{1}{n}+1}\right)}{\ln\frac{R_{1}}{R_{2}}}\right]$$

In Equation (14), the position of the maximum value of the runner velocity is $r=r_{m}$:(15)$$r_{m}=\sqrt[{\frac{1}{n}+1}]{\frac{n\left({R_{1}}^{\frac{1}{n}+1}-{R_{2}}^{\frac{1}{n}+1}\right)}{(n+1)\ln\frac{R_{1}}{R_{2}}}}$$

And the maximum velocity is as follows:(16)$$u_{2\max}=\frac{n}{n+1}{\left[\frac{\left(\Delta P_{2}+\rho_{2}gL_{2}\right)}{2KL_{2}}\right]}^{\frac{1}{n}}\left[{R_{1}}^{\frac{1}{n}+1}-{r_{m}}^{\frac{1}{n}+1}+\frac{\ln\frac{r_{m}}{R_{1}}\left({R_{1}}^{\frac{1}{n}+1}-{R_{2}}^{\frac{1}{n}+1}\right)}{\ln\frac{R_{1}}{R_{2}}}\right]$$

Substituting Equation (14) back to Equation (12), Equation (17) can be obtained as follows:(17)$$\tau_{2}=\frac{\left(\Delta P_{2}+\rho_{2}gL_{2}\right)}{2L_{2}}{\left|r^{\frac{1}{n}}-\frac{n\left({R_{1}}^{\frac{1}{n}+1}-{R_{2}}^{\frac{1}{n}+1}\right)}{r\left(n+1\right)\ln\frac{R_{1}}{R_{2}}}\right|}^{n}$$

According to Equations (14) and (17), the theoretical stress distribution and velocity distribution in the outer runner of a coaxial nozzle are shown in Figure 1. The position of the maximum value is closer to the center of the coaxial nozzle compared to $r=(R_{1}+R_{2})/2$.

#### 2.3.2. Theoretical Analysis of Cross-linking Processes

Since the normal and tangential adhesion behavior of hydrogel materials during cross-linking is highly linear [28], a linear elastic traction separation model was used to describe the bonding behavior, as shown in Figure 1. From this, we can obtain the following equation:(18)$$t=\left[\begin{array}{c} t_{1}\\ t_{2}\\ t_{3} \end{array}\right]=\left[\begin{array}{c} K_{11}\\ K_{21}\\ K_{31} \end{array}\begin{array}{c} {\;K}_{12\;}\\ \;K_{22}\\ {\;K}_{32} \end{array}\;\begin{array}{c} K_{13}\\ K_{23}\\ K_{33} \end{array}\right]\left[\begin{array}{c} \delta_{1}\\ \delta_{2}\\ \delta_{3} \end{array}\right]=K\delta$$
where t is the nominal traction vector, and its three components are $\left(t_{1},t_{2}{,t}_{3}\right)$, where $t_{1}$ is the forward traction, and $t_{2}$ and $t_{3}$ are the tangential traction. δ is the corresponding separation vector, and its three components are $\left(\delta_{1},\delta_{2}{,\delta }_{3}\right)$, respectively. The nine elements in *K* denote the direction vectors in different directions.

In general, the tangential and normal components of stiffness are not coupled. That is to say, simple normal stress does not generate cohesion in the shear direction, and simple shear slip does not generate any cohesion in the normal direction. Therefore, it is necessary to define $K_{11},K_{22}{,K}_{33}$, and the values of the other six elements of the stiffness matrix are all zero.
(19)$$K_{12}=K_{13}=K_{21}=K_{23}=K_{32}=K_{31}$$

In the online elastic traction separation model, the stiffness in the three directions can be defined as follows:(20)$$\left\{\begin{array}{c} K_{11}=t_{1}^{t}/\delta_{1}^{0}\\ K_{22}=t_{2}^{t}/\delta_{2}^{0}\\ K_{33}=t_{3}^{t}/\delta_{3}^{0} \end{array}\right.$$
where $t_{1}^{t}$ is the maximum normal separation force, $t_{2}^{t}$ and $t_{3}^{t}$ are the maximum tangential separation forces, $\delta_{1}^{0}$ is the positive initial separation displacement, and $\delta_{2}^{0}$ and $\delta_{3}^{0}$ are the tangential initial separation displacements. In addition, for isotropic materials, there is $\;K_{22}=K_{33}$.

#### 2.3.3. Theoretical Analysis of Coaxial Jet Molding Process

During the extrusion process of manufacturing vascular scaffolds, we can see that the deposition rate of gel is constantly changing. In the beginning, the gel after crosslinking is deposited slowly and their tubular diameters are large. Under the effect of gravity, the deposition rate of calcium alginate gel is increasing, and will eventually maintain a stable speed and diameter. During the flow process, the inner and outer diameters of the extrusion duct are always inconsistent with those of the coaxial nozzle, where $D_{NO}$ and ${\;D}_{NI\;}$ are the inner diameters of the coaxial nozzle’s outer nozzle and the inner nozzle’s inner diameter, respectively. $D_{O}$ and $D_{I}$ are the outer and inner diameters of the extrusion duct, respectively. $V_{DI}$ is the velocity of sodium alginate at the instant of flow from the nozzle, and $V_{DE}$ is the calcium alginate deposition velocity, as shown in Figure 1. According to Equation (21), the runner flow rate for internal and external flow can be found.
(21)$$Q=\int_{r_{\min}}^{r_{\max}}2\pi rudr$$

Substituting the flow velocity functions and boundary conditions for the inner and outer flow runners in Section 2.3.1, the runner flow rate for internal and external flow $Q_{1}$ and $Q_{2}$ can be obtained as follows:(22)$$Q_{1}=\int_{0}^{R}2\pi ru_{1}dr=\frac{\left(\Delta P_{1}+\rho_{1}gL_{1}\right)\pi R^{4}}{8\mu_{1}L_{1}}$$
(23)$$\begin{array}{c} Q_{2}=\int_{R_{2}}^{R_{1}}2\pi ru_{2}dr\\ =\frac{2\pi n}{n\;+\;1}{\left[\frac{\left(\Delta P_{2}\;+\;\rho_{2}gL_{2}\right)}{2KL_{2}}\right]}^{\frac{1}{n}}\left[\frac{n\;+\;1}{6n\;+\;2}\left({R_{1}}^{\frac{1}{n}+3}-{R_{2}}^{\frac{1}{n}+3}\right)-\frac{\left({R_{1}}^{2}\;-\;{R_{2}}^{2}\right)\left({R_{1}}^{\frac{1}{n}+1}\;-\;{R_{2}}^{\frac{1}{n}+1}\right)}{4\ln\frac{R_{1}}{R_{2}}}\right] \end{array}$$

Assuming that the duct has a uniform diameter and a uniform deposition velocity ($V_{DE}$), and assuming that the volume of boink does not change much before and after cross-linking, it can be regarded as an incompressible fluid and, therefore, the flow rate can be calculated as follows:(24)$$Q_{2}=\pi \left(\frac{D_{NO}^{2}-D_{NI}^{2}}{4}\right)V_{DI}=\pi \left(\frac{D_{O}^{2}-D_{I}^{2}}{4}\right)V_{DE}$$

Then, the deposition velocity ($V_{DE}$) can be expressed as follows:(25)$$V_{DE}=\frac{4Q_{2}}{\pi \left(D_{O}^{2}-D_{I}^{2}\right)}$$

#### 2.3.4. Numerical Simulation of Coaxial Jet

Fluid simulation, solution, and computation of the flow process of bioink in the flow channel of the coaxial nozzle were carried out using ANSYS Fluent 2022 R1 software. The internal and external runner fluids of the coaxial nozzle were simplified using SolidWorks 2022 software to create the 3D model. Adaptive sizing was used to control the volume size and achieve precise structural meshing after conducting mesh analysis, as shown in Figure 2c. Based on theoretical modeling analysis, laminar flow was chosen as the flow state. New fluids were created in the Fluent material library by specifying properties such as density and dynamic viscosity. The inlet and outlet boundary conditions were set to velocity and pressure, respectively. Then, the wall was set to be non-slip and impermeable. Different discretization methods were applied to the control equations. The first-order upwind approach was used to discretize the momentum equations. For pressure-velocity coupling, the Pressure Implicit Split Operator (PISO) method was used.

The cloud diagram in Figure 3a illustrates the overall pressure distribution of the flow channel axial section. Due to the horizontal axis of the outer runner and the high viscosity of the bioink, the overall pressure in the outer runner is significantly higher than that in the inner runner. This creates pressure extreme at the inlet of the outer runner. Therefore, attention needs to be paid to the inlet pressure after the mixed-cell bioink flows into the syringe. Too much inlet pressure can result in reduced cell survival. From the axial pressure distribution curve of the outer runner, as shown in Figure 3b, we can learn that as the feed speed increases, the overall pressure of the outer runner increases.

The cloud diagram in Figure 3c illustrates the overall velocity distribution of the flow channel axial section. At the outlet, the flow channel narrows, causing the outlet velocity to increase due to the same flow rate between the inlet and outlet. The highest velocity value is found near the axis of the outer runner’s horizontal thin runner. From the axial velocity distribution curve of the inner runner, as shown in Figure 3d, we can learn that as the feed speed increases, the inner runner velocity increases. From the radial velocity distribution curve of the runner, as shown in Figure 3f, it can be observed that the position of the maximum value is closer to the center of the circle compared to the middle of the inner and outer diameter, in the outer runner. And the velocity distribution of the inner runner follows a nearly quadratic pattern, which is consistent with the results of the theoretical modeling calculations in Section 2.3.1. Similarly, the center flow velocity is higher than the edge flow velocity at the outlet of the runners, as shown in Figure 3e. Therefore, we have to adjust the inlet speed appropriately, which could affect the extrusion effect.

The cloud diagram in Figure 3g shows the local wall shear stress distribution of the outer runner axial section. There are significant shear stresses near the walls of the outer runners, and the extreme values are located at the horizontal–vertical junction. From the local radial wall shear stress distribution curve of the outer runner, as shown in Figure 3h, we can learn that as the feed speed increases, the shear stress increases. The cell deaths encountered during the printing process are often caused by the shear force on the cells.

Combined with the above coaxial flow channel numerical simulation analysis, when the feed speed increases, the stress in the flow channel as well as the bioink extrusion speed increases correspondingly. However, too high stress and speed are not favorable for cell survival and keeping the uniformity of the hydrogel tube diameter. At the same time, when the feed speed is too low, the hydrogel cannot form a tubular structure. In summary, the preliminary selection of the feed speed is 2 mm/s.

Based on the coaxial jet runner numerical simulation discussed above, the numerical simulation of the bioink flow from the nozzle into the air section was continued. Due to the presence of more than two immiscible fluids in the domain, the multiphase Volume of Fluid (VOF) model was used to solve the momentum equations and track the volume fractions of all Eulerian phases. It consisted of three phases: the primary phase is air, while the secondary phases are bioink and cross-linking agent.

The two fluidic materials, cross-linking agent and bioink, flow from the nozzle into the air in the process state shown in Figure 4. At the moments of *t* = 0.5 s, 1 s, and 1.5 s, as shown in Figure 4a, the bioink with the symmetrical shape is uniformly flowing out from the coaxial nozzle, exhibiting a continuous linear form. As depicted in Figure 4b, the bioink is observed to flow out while encapsulating the cross-linking agent. Furthermore, the radial section reveals that the outer fluid section is approximately annular, and indicates that the fluid flowing out of the outer channel forms a tubular shape.

According to the above numerical simulation results, attention should be paid to controlling the flow velocity of the cross-linking agent in the inner flow channel, which should be preceded by the flow of the bioink in the outer flow channel to ensure that the two are in full contact with the reaction. However, excessive flow rates should be avoided to prevent compromising the uniformity of the formed structure. At the same time, the distance between the nozzle of the coaxial nozzle and the substrate should be optimized

### 2.4. Theoretical Analysis and Numerical Simulation of Vascular Scaffolds

The state of blood flow through a blood vessel varies depending on the diameter of the vessel. Many current studies on blood flow dynamics, in most cases, consider blood as an incompressible Newtonian fluid. Therefore, when conducting hemodynamic studies, we make certain assumptions about certain parameters:It is assumed that the blood flow velocity remains constant in the blood vessels;It is assumed that the deformation of the vessel wall is minimal as blood flows through the vessel;It is assumed that blood is an incompressible Newtonian fluid.

#### 2.4.1. Theoretical Analysis of Vascular Scaffolds

During blood flow, a dimensionless parameter can be used to describe the relationship between the magnitude of the blood viscous force and the local inertia force. This parameter is known as the frequency parameter. It is defined as follows:(26)$$\alpha =\frac{D}{2}\sqrt{\frac{\omega }{V}}$$
where ω, *V* and *D* represent the circular frequency of blood flow, velocity of the blood, and inner radius of blood vessels, respectively.

When blood flows through the arteries at a certain velocity, the viscosity of the blood and the characteristic impedance of the vascular segment cause a decrease in blood pressure. The amount of decrease in blood pressure depends on the level of impedance in the vascular segment. The higher the impedance, the greater the decrease in blood pressure, and conversely, the lower the impedance, the smaller the decrease.

The equation describing the characteristic impedance of a blood vessel is as follows:(27)$$Z_{c}=\frac{\rho a}{A}$$
where $Z_{c}$, *A*, ρ and *a* are the vascular characteristic impedance, vascular segment cross-sectional area, blood density, and pulse wave velocity, respectively.

The blood pressure drop equation is as follows:(28)$$∆P=Z_{c}q$$
where Δ*P*, *u* and *q* represent the pressure drop, axial blood flow velocity, and flow rate, respectively, and *q* = *Au*, which is obtained by joining the above two equations:(29)∆P=ρau

The continuity equation for the flow of blood through the vascular network [29,30,31], assuming that the flowing blood is an incompressible Newtonian fluid, is given by the following:(30)$$\frac{\partial u}{\partial x}+\frac{1}{r}\frac{\partial \left(rv\right)}{\partial r}=0$$

The blood movement equation is as follows:(31)$$\left\{\begin{array}{c} \frac{\partial v}{\partial t}+u\frac{\partial v}{\partial x}+v\frac{\partial v}{\partial r}=-\frac{1}{\rho }\frac{\partial P}{\partial r}+\upsilon \left(\frac{\partial^{2}v}{\partial r^{2}}+\frac{1}{r}\frac{\partial v}{\partial r}+\frac{\partial^{2}v}{\partial x^{2}}-\frac{v}{r^{2}}\right)\\ \frac{\partial u}{\partial t}+u\frac{\partial u}{\partial x}+v\frac{\partial u}{\partial r}=-\frac{1}{\rho }\frac{\partial P}{\partial x}+\upsilon \left(\frac{\partial^{2}u}{\partial r^{2}}+\frac{1}{r}\frac{\partial u}{\partial r}+\frac{\partial^{2}u}{\partial x^{2}}\right) \end{array}\right.$$
where *u* is the axial blood flow velocity, *v* is the radial blood flow velocity, *P* is the blood pressure, and υ is the blood kinematic viscosity.

Since the radial displacement of the blood is much smaller than the axial displacement of the blood during the flow process, we can ignore some higher-order inertial and viscous terms of the above equation, and thus the simplified equation obtained is as follows:(32)$$\left\{\begin{array}{c} \frac{\partial u}{\partial x}+\frac{1}{r}\frac{\partial \left(rv\right)}{\partial r}=0\\ \frac{\partial u}{\partial t}=-\frac{1}{\rho }\frac{\partial P}{\partial x}+\upsilon \left(\frac{\partial^{2}u}{\partial r^{2}}+\frac{1}{r}\frac{\partial u}{\partial r}\right)\\ \frac{\partial P}{\partial r}=0 \end{array}\right.$$

During the flow of blood, pressure is exerted on the walls of the blood vessels and hence the axial and radial forces on the vessel walls are as follows:(33)$$\left\{\begin{array}{c} \rho_{\omega }\frac{\partial^{2}\xi }{\partial t^{2}}=\frac{P\;-\;P_{e}}{H}-\frac{h}{H}\frac{E}{1\;-\;\mu^{2}}\left[\frac{\mu }{R\left(x\right)}\frac{\partial \zeta }{\partial x}+\frac{\xi }{R^{2}\left(x\right)}\right]-\frac{2\eta }{H}{\left(\frac{\partial v}{\partial r}\right)}_{r=R}\\ \rho_{\omega }\left(\frac{\partial^{2}\zeta }{\partial t^{2}}+\omega_{0}\zeta \right)=\frac{h}{H}\frac{E}{1\;-\;\mu^{2}}\left[\frac{\partial^{2}\zeta }{\partial x^{2}}+\frac{\mu }{R\left(x\right)}\frac{\partial \xi }{\partial x}\right]-\frac{\eta }{H}{\left(\frac{\partial u}{\partial r}+\frac{\partial v}{\partial x}\right)}_{r=R} \end{array}\right.$$
where ξ is the displacement of the vessel wall along the axial direction, ζ is the displacement of the vessel wall along the radial direction, *P* and $P_{e}$ are the internal and external pressures exerted on the vessel wall, respectively, *H* is the actual thickness of the vessel wall, $\rho_{\omega }$ is the density of the vessel wall, *h* is the theoretical thickness of the vessel wall, μ is the Poisson’s ratio, *R* is the radius of the vessel wall, and *E* is the elastic modulus.

If the interaction force between the tissues of the vessel wall is neglected, then *H* = *h*. If *h* ≪ *R*, the system of equations can be simplified as follows:(34)$$\left\{\begin{array}{c} \frac{P\;-\;P_{e}}{H}-\frac{E}{1\;-\;\mu^{2}}\frac{\xi }{R^{2}}=0\\ \rho_{\omega }\frac{\partial^{2}\zeta }{\partial t^{2}}=\frac{E}{1\;-\;\mu^{2}}\left[\frac{\mu }{R\left(x\right)}\frac{\partial \xi }{\partial x}+\frac{\partial^{2}\zeta }{\partial x^{2}}\right]-\frac{\eta }{H}{\left(\frac{\partial u}{\partial r}\right)}_{r=R} \end{array}\right.$$

In the direction of the vessel wall *r* = *R*(*x*), the state of motion between the blood and the vessel wall is coupled, and the coupling conditions are as follows:

Boundary conditions:(35)$$\frac{\partial \zeta \left(x,t\right)}{\partial t}=u{\left[x,R\left(x\right),t\right]}_{r=R_{0}}$$
(36)$$\frac{\partial \xi \left(x,t\right)}{\partial t}=v\left[x,R\left(x\right),t\right]$$

Wall coupling conditions in blood vessels:(37)$$v{\left[x,R\left(x\right),t\right]}_{r=R_{0}}=0$$
(38)$${\frac{\partial v\left[x,R\left(x\right),t\right]}{\partial r}}_{r=R_{0}}=0$$

The system of Equation (33) shows that the expression for the elastic deformation of the vessel wall occurs as follows:(39)$$\xi =\frac{\left(1-\mu^{2}\right)R^{2}}{hE}\left(P-P_{e}\right)$$

From the generalized Hooke’s law, the stress on the vessel wall during elastic deformation is obtained as follows:(40)$$\sigma_{\theta }=E\varepsilon_{\theta }=E\frac{\xi }{R}$$

Equation (41) can be found by associating Equations (39) and (40):(41)$$\sigma_{\theta }=\frac{\left(1-\mu^{2}\right)R}{h}\left(P-P_{e}\right)$$

#### 2.4.2. Numerical Simulation of Vascular Scaffolds

ANSYS Fluent 2022 R1 software was used for fluid simulation, solution, and calculation of the blood flow process in human arterial vessels. SolidWorks 2022 software was utilized to construct the 3D model of the vascular network, which was then imported into ANSYS 2022 R1 for meshing, as shown in Figure 5c. Since the majority of blood flow in the human body is laminar, the fluid flow state is referred to as laminar flow. According to the attributes of human blood density and dynamic viscosity, the new fluid was added to the Fluent material library. According to the average value of the human heartbeat cycle pressure, the default working pressure value for the simulation was changed to 124 mmHg. The rest of the solution calculation method was set up in the same way as in Section 2.3.4.

The pressure distribution cloud diagram of the axial section of the vascular network is shown in Figure 5d. The diagram illustrates that the pressure is highest at the entrance of the vascular network and gradually decreases as the vessels bifurcate at all levels, reaching its minimum at the exit of the vascular network. From the pressure distribution curve of the tributary vascular, as shown in Figure 5e, we can learn that the pressure drop gradient in the main blood vessel is greater than that in branch blood vessels.

The axial sectional velocity distribution cloud diagram of the vascular network is shown in Figure 5f. The blood flow velocity is approximately symmetrically distributed within the vascular network. Near the vessel walls, the flow velocity is relatively low, forming a boundary layer. Conversely, in the central region of the vessel, the flow velocity is higher. This inhomogeneous distribution of flow velocities is attributed to the combined effects of frictional resistance between the blood and vessel walls, as well as the viscous forces within the blood itself. Figure 5g illustrates the changes in blood flow velocity distribution within the tributary vessels. The velocity gradually decreases after bifurcation at all levels and then increases again when the blood converges into the main vessel.

Based on the above numerical simulation of human blood flow, the numerical simulation of solid–liquid coupling in human blood vessels was continued. The inlet of the vascular network to the first level of bifurcation as the highest pressure section was selected and local coupling simulation was performed. The static pressure in blood flow was introduced and applied to the inner surface of the blood vessel. Subsequently, the inner surface was designated as the solid–liquid coupling contact surface. The vessel’s port section was set up as remote displacement to restrict movement in six degrees of freedom.

The total deformation cloud of the blood vessel is shown in Figure 6b. In the straight-through portion of the vessel, the deformation of the vessel wall in the radial direction arises. Due to the pressure gradient drop in the vessel, the radial displacement of the vessel wall gradually decreases from the entrance along the axial direction. Due to the local high pressure caused by blood flow impacting the vessel wall at bifurcations, the maximum deformation occurs at the vascular bifurcation.

The equivalent stress distribution in the vessel axial section is depicted in Figure 6c. In the straight-through section of the vessel, the stress decreases gradually from the inner wall to the outer wall of the blood vessel along the radial direction, exhibiting a laminar distribution. At the bifurcation of the vessel, the stress distribution was negatively correlated with the deformation, and the highest stress value was observed on the inner surface of the bifurcated vessel wall. Moreover, the distribution of the equivalent elastic strain in the vessel is generally consistent with the distribution of equivalent stress, as shown in Figure 6d.

### 2.5. Fabrication Methods

#### 2.5.1. 3D Bioprinting System Based on Coaxial Jet

The experimental platform construction was based on coaxial 3D bioprinting, and the schematic and physical drawings are shown in Figure 7a,c. The experimental equipment required for the fabrication of vascular scaffolds mainly included the following: coaxial nozzle (Ispin, Hefei, China), micro syringe pump (LongerPump, Baoding, China), PC, filming device, sodium alginate, calcium chloride, viscometer (LICHEN, Changsha, China), HUVECs, and so on. Coaxial nozzle (14G–18G) dimensions are shown in Table 1. Figure 7b illustrates the micro syringe pump (LSP04-1A), which is operated by initially drawing the bioink and cross-linking agent into two syringes. Subsequently, the bioink and cross-linking agent were delivered to the inner and outer nozzles through two separate tubes. The extrusion speed was controlled by adjusting the black slider to regulate the feed speed. The device boasts the advantage of concurrent operation of up to four working channels, alongside its compatibility with syringes of varying specifications, enabling precise flow rate control.

In the preliminary selection, 2.5% sodium alginate solution and 2% calcium chloride solution were filled into 2 syringes, respectively. The coaxial nozzle assembly consisted of a feed tube (syringe) and inner and outer nozzles. The sodium alginate solution entered the outer tube portion, while the calcium chloride cross-linking agent entered the inner tube portion. When the two solutions came into contact, the bioink and the cross-linking agent began to cross-link. By setting the parameters of the injection pump, the feed speed of the internal and external nozzles was not higher than 2 mm/s, and the syringe connected to the inner nozzle drew in a greater amount of calcium chloride solution compared to the sodium alginate connected to the outer nozzle. This ensured that the black slider made contact with the syringe filled with calcium chloride solution and pushed it forward before coming into contact with the sodium alginate solution. As a result, the calcium chloride solution was extruded from the nozzle first.

#### 2.5.2. Mold System

The mold set for bifurcated vascular structures comprised four molds: two concave molds and two convex molds. The side edges of the concave and convex parts of the mold designed in SolidWorks had a concentric circular arc shape with larger diameters (2 mm and 1.5 mm). The molds were 3D-printed using ABS material and subsequently polished for smooth demolding.

The microcasting method involved the following specific steps, as shown in Figure 8: First, mold 1 and mold 2 were fitted together using concave–convex positioning blocks to align the central axes of the two molds. Then, low-concentration calcium chloride cross-linked alginate hydrogel was injected into the mold entrance, and allowed to set for ten minutes at room temperature. The above steps were repeated, and another pair of concave–convex molds was used to prepare the other side of the bifurcated structure. After that, the set of two concave molds was axially fitted together, and calcium chloride solution was injected into the entrance. The molds were allowed to be set for ten minutes to allow the two structures to fully cross-link and adhere. Finally, the molds were slowly demolded to obtain a bifurcated structure of hydrogel tubes.

### 2.6. Experimental Verification

#### 2.6.1. Degradability Test

A vascular scaffold fabricated with 3% sodium alginate solution and 2.5% calcium chloride solution was selected and adequate dryness was ensured. Subsequently, the weight of the scaffold was denoted as *w_i_* (initial weight). The scaffold was then submerged in simulated body fluid (SBF) (Shanghai Aladdin Biochemical Technology Co., Ltd., Shanghai, China) and placed in a 37 °C incubator with 5% carbon dioxide. Over the course of the 14-day experimental period, the scaffold was periodically removed from the incubator every two days. It was then blotted dry to eliminate excess moisture and reweighed, recorded as *w_r_* (residual weight). The degradation rate was calculated using the following equation:(42)$$Degradation\;rate\;\left(\%\right)=\frac{w_{i}-w_{r}}{w_{i}}\times 100\%$$

#### 2.6.2. Water Absorption Test

Different concentrations of sodium alginate solutions were selected for comparison and experiments using the 2.5% calcium chloride solution as a cross-linking agent. The scaffold was placed in a dry environment at room temperature for 4 days to allow for dehydration. Afterward, it was weighed and recorded as *w_d_* (dry weight). Then, it was placed in PBS (Shanghai Aladdin Biochemical Technology Co., Ltd., Shanghai, China) solution and weighed on an analytical balance hourly and recorded as *w_w_* (wet weight). After 8 h of treatment, the above experiments were repeated every two hours until the basic value of the scaffold mass remained unchanged.
(43)$$Water\;absorption\;rate\;\left(\%\right)=\frac{w_{w}-w_{d}}{w_{d}}\times 100\%$$

#### 2.6.3. Mechanical Properties Test

The vascular scaffolds, made of sodium alginate with concentrations of 2% and 2.5%, and calcium chloride with concentrations of 2%, 3%, and 4%, were subjected to tensile experiments, respectively. The tensile strength of the scaffolds was tested using a WDX-100 electronic universal material testing machine (Shanghai Gehong Instrument Co., Ltd., Shanghai, China). Based on the stress σ and strain ε experienced by the vascular scaffold, the elastic modulus *E* was derived through Hooke’s law, and the ultimate stress $\sigma_{us}$ was also determined during the testing process.
(44)$$E=\frac{\sigma }{\varepsilon }$$
(45)$$\sigma_{us}=\frac{4F}{\pi \left(D^{2}-d^{2}\right)}$$

#### 2.6.4. Biocompatibility Test

Endothelial cells were stained with Calcein-AM (Shenggong Biotechnology (Shanghai) Co., Ltd., Shanghai, China) and cultivated for a period of time. Observations of cell proliferation and distribution were conducted under a fluorescent inverted microscope (Shanghai Dianying Optical Instrument Co., Ltd., Shanghai, China) at five time points: 12 h, 1 day, 3 days, 5 days, and 7 days. Using ImageJ2 software, a statistical analysis was conducted to count the living cells within the field of view at various time points and under the same magnification.

## 3. Results and Discussion

### 3.1. Experimental Analysis of Fabrication Process

#### 3.1.1. Printability of Vascular Scaffolds and Optimization of Process Parameters

It can be seen from Figure 9a that the extrudate had a linear shape. Observed under a high-definition microscope (Shanghai Dianying Optical Instrument Co., Ltd., Shanghai, China), the outer surface layer of the scaffold in the air exhibited a gelatinous texture and gloss. In a calcium chloride solution, the inner and outer diameters of the gel tube were clearly observable. In order to determine if the hydrogel extrudate had cross-linked into a tube, red ink was injected into the syringe and connected to the extrudate. A total of 5 s later, the hydrogel was full of red ink. It can be seen that the sodium alginate solution and calcium chloride cross-linking agent underwent a cross-linking reaction, resulting in a tubular extrudate, as shown in Figure 9b.

In order to obtain the optimal concentration ratio of bioink and cross-linking agent for coaxial printing of vascular scaffolds, different concentration gradient combinations were selected for multiple printing experiments. The experimental results were correlated with the printability of the scaffold, as shown in Figure 9c. The gelation process depended on diffusion and occurred on the entire cross-section of the duct. When the concentrations of bioink and cross-linking agent were too low, the cross-linking phenomenon was not significant. When the concentration of bioink was too high, the printing needle was easy to be blocked due to the high viscosity. When the concentration of the cross-linking agent was too high, the excessive calcium source made the gel rate too fast, which blocked the subsequent cross-linking reaction. In summary, when the concentration of sodium alginate solution was about 3% and the concentration of calcium chloride solution was about 2.5%, it was easier to print a scaffold with a completely symmetrical tubular structure that was more transparent.

Statistical data were collected on the diameters of scaffolds printed using combinations of 2.5% calcium chloride with various concentrations of sodium alginate, and 3% sodium alginate with different concentrations of calcium chloride, as shown in Figure 9d,e. The results showed that the effect of material concentration on the diameter size had no obvious trend. However, when the flow rate of the cross-linker remained stable (1.0 mL/min), Figure 9f shows that with the increase in the flow rate of sodium alginate solution during coaxial printing, the diameter of the scaffold showed an increasing trend. This can be explained by the increased diffusion rate of the alginate network and the higher radial force exerted on the catheter wall. According to the above experimental results, although smaller diameter catheters can be fabricated, the results may damage the mechanical integrity of the catheter, resulting in uneven ducts. Figure 9g shows the inner and outer diameter ratios of the ducts fabricated with 3% alginate and 2.5% calcium chloride at different flow rates. We conducted comparative tests with the flow rates of sodium alginate at 0.4 mL/min and cross-linking agent at 0.8 mL/min, 1.6 mL/min, 2.4 mL/min, and 3.2 mL/min. We found that with the increase in the flow rate of the cross-linking agent to sodium alginate, the inner and outer diameter ratios of the tubes gradually increased. And the increase in the absolute flow rates of both fluids did not affect this trend. This would result in a smaller wall thickness of the gel tube, causing a decrease in the mechanical performance of the vascular scaffold. In summary, when printing, the flow rate of sodium alginate solution should not be less than 0.5 mL/min, and the flow rate ratio of cross-linking agent to bioink should be about 2.

#### 3.1.2. Research of Vascular Network Scaffolds

Based on the single-through structures fabricated by coaxial printing and the bifurcated structures fabricated by the mold method, we connected the Y-shaped and straight tubes end-to-end. First, we modified both types of gel tubes to obtain multilevel bifurcated structures, as shown in Figure 10a–c. Then, we integrated them into each other. To successfully adhere the hydrogel tubes to the bifurcated tubular structures, the outer diameter of the hydrogel tubular structure had to be smaller than the inner diameter of the bifurcated tubular structure. Subsequently, a small amount of 5% sodium alginate solution was added dropwise to the outer gap of the spliced site, followed by crosslinking with 3% calcium chloride solution to achieve tight adhesion. Finally, we repeated the above process to obtain vascular scaffolds with vascular network structures, as shown in Figure 10f.

We performed simple perfusion experiments on the obtained vascular network structures and the results indicated that the vascular scaffolds could complete basic liquid transport function. The initial perfusion flow rate was set at 0.5 mL/s, with subsequent increments of 0.1 mL/s each time. However, as the perfusion flow rate increased, reaching about 2 mL/s, a higher internal pressure resulted in minor leakage of liquid at the connection site.

### 3.2. Characterization of Vascular Scaffolds

#### 3.2.1. Degradability of Vascular Scaffolds

The in vitro simulation of the degradation of biological scaffold materials can indirectly reflect the anti-fluid corrosion and deformation properties of the scaffold, as well as the intensity of degradation and metabolism after implantation of the cell-active construct into the body. This is an important property of biological scaffold materials. Vascular scaffolds should have an appropriate degradation rate that allows sufficient time for endothelial cells to proliferate while avoiding occupying the growth space of the extra-cellular matrix for extended periods of time.

It can be seen from Figure 11a that the vascular scaffold exhibited a relatively stable degradation rate during the experimental period, and the degradation rate after 14 days was 34.8%. The experimental results indicated that the fabricated vascular scaffold can provide mechanical support for the attachment and proliferation of endothelial cells, and possesses degradable capabilities.

#### 3.2.2. Water Absorption of Vascular Scaffolds

Vascular scaffolds should possess a moderate water absorption capacity to mimic natural blood vessel function. Excessively low water absorption rates might potentially hinder physiological function, while excessively high rates could lead to issues such as vessel expansion, mechanical instability, and increased thrombus formation risk. Therefore, selecting an appropriate water absorption rate is crucial.

It can be seen from Figure 11b that the maximum water absorption decreased as the concentration of alginate solution increased. This is because the cross-linking reaction may not be sufficient in lower concentrations of alginate solution. Increasing the concentration of alginate solution can promote the cross-linking reaction and form a more compact network structure, which limits the diffusion and penetration of water molecules in the gel, leading to a decrease in the water absorption capacity of the gel. The group with a concentration of 3% exhibited the strongest water absorption capacity, with a maximum water absorption rate of 660%. The 5% alginate solution group had the weakest water absorption capacity, with a maximum water absorption rate of 280%. The maximum water absorption saturation time decreased with increasing concentration. The water absorption curve of the 3% group flattened after 48 h, while the water absorption curve of the 5% group flattened after 4 h.

#### 3.2.3. Mechanical Properties of Vascular Scaffolds

Since the blood vessel wall in the human body is subjected to the pressure of blood during blood flow and is affected by the cyclicity of the heartbeat, vascular scaffolds must meet certain mechanical requirements.

As shown in Figure 11c, the ultimate stress of the tested vascular scaffolds in all groups exceeds the maximum stress of the vessel wall obtained from the partially coupled simulation in Section 2.4.2, suggesting that their mechanical properties meet the requirements of the simulation outcomes. In the tensile test, the higher concentrations of sodium alginate and calcium chloride led to the formation of a denser network structure through ionic crosslinking, thereby exhibiting better mechanical properties. The SA2Ca2 in Figure 11c represents a scaffold cross-linked by a solution of sodium alginate with a concentration of 2% and calcium chloride with a concentration of 2%. From the figure, it is evident that as the concentration of sodium alginate and cross-linking agent increased, both the ultimate stress and elastic modulus of the vascular scaffold also increased. The ultimate stress of the scaffold cross-linked with 2.5% sodium alginate and 4% calcium chloride reached 1.88 MPa, and the elastic modulus reached 2 MPa. On the other hand, the ultimate stress of the scaffold cross-linked with 2% sodium alginate and 2% calcium chloride was 0.81 MPa, and the elastic modulus was 0.7 MPa.

#### 3.2.4. Biocompatibility of Vascular Scaffolds

The cellular proliferative capacity of vascular scaffolds is intricately linked to their biocompatibility. As shown in Figure 11d, after being cultured in a cell incubator for 12 h, the endothelial cells within the vascular scaffold commenced a rapid proliferative phase, with a steady increase in the number of living cells over the next three days. However, after five days, the growth trend of living cells decelerated due to the compression of growth space within the vascular scaffold. With the passage of time, the cells in the wall portion of the vascular scaffold grew and multiplied rapidly, completely filling the wall with cells by the seventh day. The data from the cell viability assay showed that the cells were biocompatible with the vascular scaffold.

According to the theoretical analysis in Section 2.3.1, the cell viability is expected to be the lowest near the wall surface of the duct. Conversely, it should be the highest in the middle of the hydrogel, which corresponds to the point where shear stress disappears inside the coaxial nozzle. This result was consistent with the observation in the figure that dead cells were primarily located along the inner and outer walls of the duct. In addition, the number of dead cells in the inner wall was greater than that in the outer wall.

## 4. Conclusions

A method for fabricating vascularized tissue based on 3D bioprinting technology that combines coaxial printing with the mold method is proposed. Bifurcation structures were fabricated using the mold method exclusively. Subsequently, single-pass hydrogel tubes were created through coaxial printing, obtaining the incorporation of complex structural vascular networks through embedding integration. Sodium alginate solution was chosen as the bioink material, and calcium chloride solution was used as the cross-linking agent. Based on the power law fluid suspension model, we formulated the theoretical model of coaxial jetting. We conducted mathematical modeling and finite element simulation to analyze the flow behavior and extrusion process of bioink. Because the distribution of viscous shear stress within the outflow channel significantly influences the survival and distribution of endothelial cells within the vascular scaffold, we subsequently formulated the theoretical model of the vascular network and conducted a numerical simulation analysis of the flow state of blood in the vascular network and the deformation of the vessel wall. And this served as a criterion for evaluating whether the mechanical properties of the vascular scaffold meet the required standards. Additionally, we roughly determined the printing parameters based on the simulation results. The results showed that when the feed speed of both the inner and outer nozzles was controlled at 2 mm/s or below, and when the concentration of sodium alginate was above 2%, a distinct tubular structure could be observed in the extruded material. Furthermore, the calcium chloride solution was extruded prior to the sodium alginate solution, allowing for cross-linking and the formation of a tube. After conducting the experimental statistics, when using a 3% sodium alginate solution in conjunction with a 2.5% calcium chloride solution, and ensuring that the flow rate of the sodium alginate was maintained at a minimum of 0.5 mL/min while the flow rate of the calcium chloride solution was approximately double that, the resultant vascular scaffold structure exhibited optimal characteristics. The degradability test results showed that the degradation rate of the vascular scaffold after 14 days was 34.8%, indicating good degradability. The water absorption test results showed that the water absorption rates of the vascular scaffolds fabricated with sodium alginate with a concentration range of 3% to 5% ranged from 280% to 660%. The ultimate stress and elastic modulus of blood vessels fabricated by a coaxial jet of 2.5% sodium alginate solution and 4% calcium chloride solution were 1.88 MPa and 2 MPa, respectively, as measured by microtensile tests. In the cellular experiments, using a fluorescent inverted microscope, the distribution and satisfactory proliferation of stained endothelial cells within the vascular scaffold were observed over a span of one week. The cells showed good biocompatibility with the vascular scaffold. Furthermore, the vascular network scaffolds successfully fabricated by integrating the coaxial jetting method with the mold method were capable of fulfilling the fundamental function of liquid transportation. However, excessively high perfusion flow rates can still result in minor leakage of liquid at the connection sites within the network structure. In summary, the fabricated vascular scaffolds possessed mechanical properties and biocompatibility compatible with human vascularized tissue, along with a networked structure, thereby initially fulfilling the fundamental requirements for implantable biological scaffolds.

However, a biomimetic vascular tissue that can be truly implanted into the human body should possess a multilayered structure composed of different cell types. Therefore, our research findings provide a method that can be used as a reference for future studies on vascularized tissue, such as utilizing a multilayer coaxial nozzle for direct printing of multilayer vascularized tissue structures, as well as strategies combining various printing methods that can be adopted to fabricate vascularized tissue with complex structures.

## Figures and Tables

**Figure 1 micromachines-15-00463-f001:**
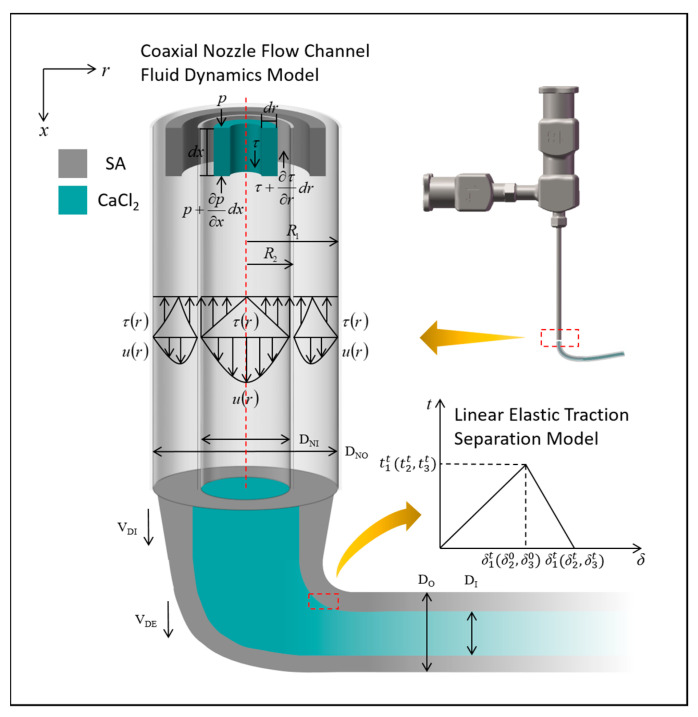
Schematic diagram of the theoretical model for each stage of the coaxial jet process.

**Figure 2 micromachines-15-00463-f002:**
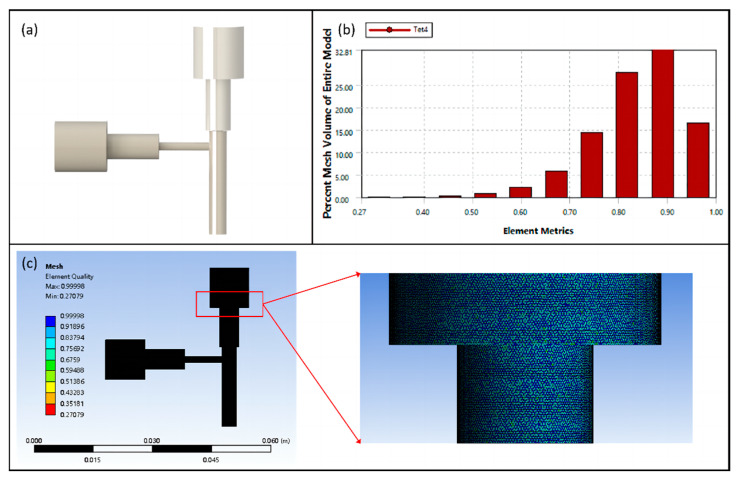
(**a**) Runner 3D model, (**b**) element quality statistics, and (**c**) overall and local meshing.

**Figure 3 micromachines-15-00463-f003:**
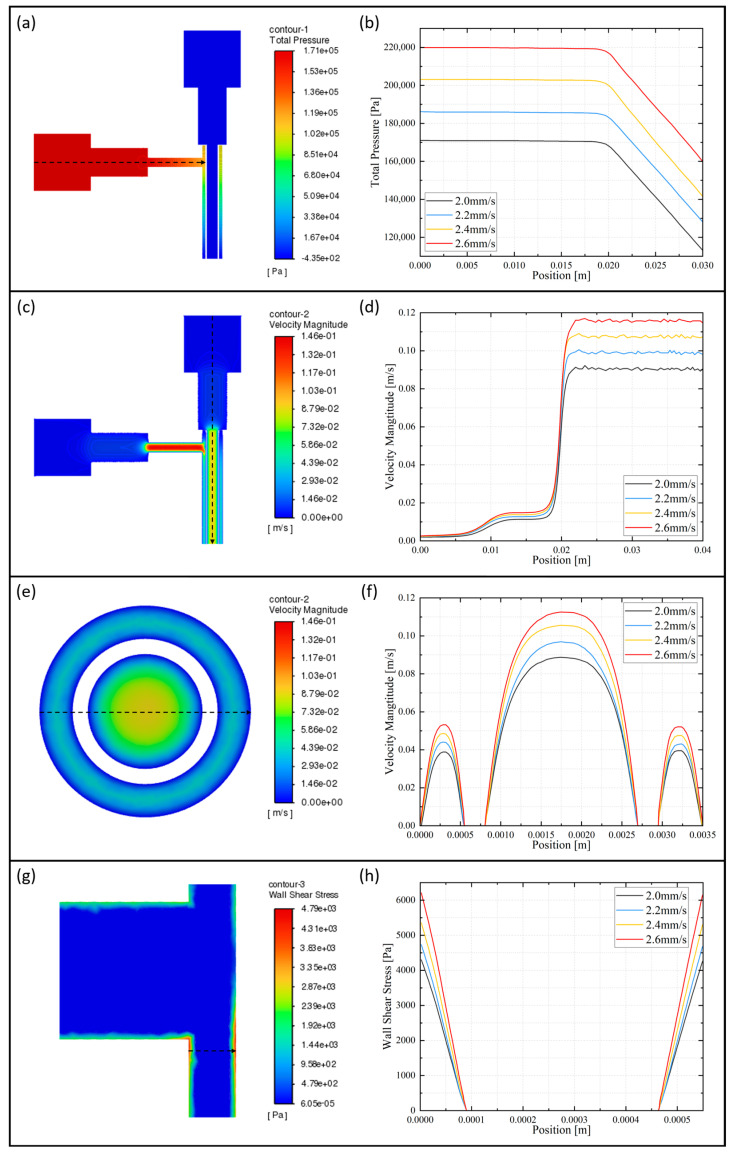
Feed speed of 2 mm/s: (**a**) runner axial section pressure overall distribution cloud; (**c**) runner axial section velocity overall distribution cloud; (**e**) runner outlet velocity cloud; and (**g**) runner axial section local wall shear stress distribution cloud. Different feed rate: (**b**) outer runner axial pressure distribution curve; (**d**) inner runner axial velocity distribution curve; (**f**) runner radial velocity distribution curve; and (**h**) outer runner local radial wall shear stress distribution curve.

**Figure 4 micromachines-15-00463-f004:**
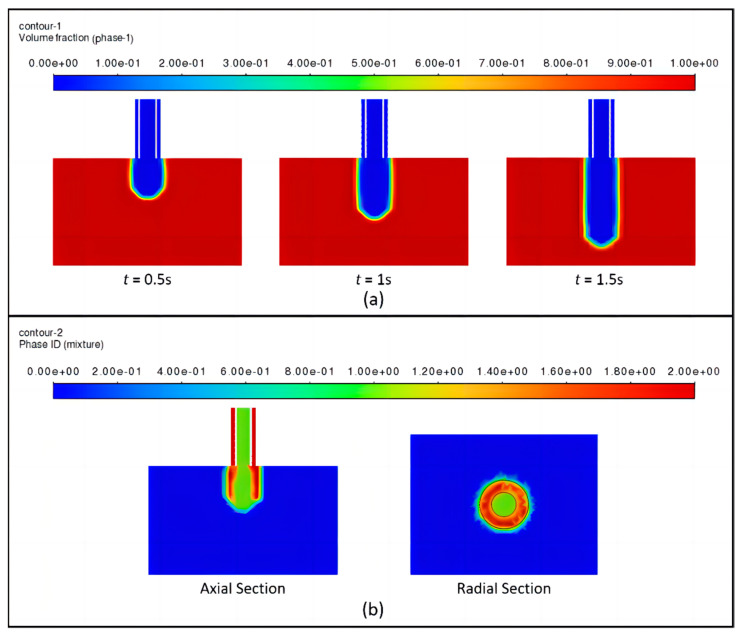
(**a**) Bioink form, (**b**) three-phase flow form.

**Figure 5 micromachines-15-00463-f005:**
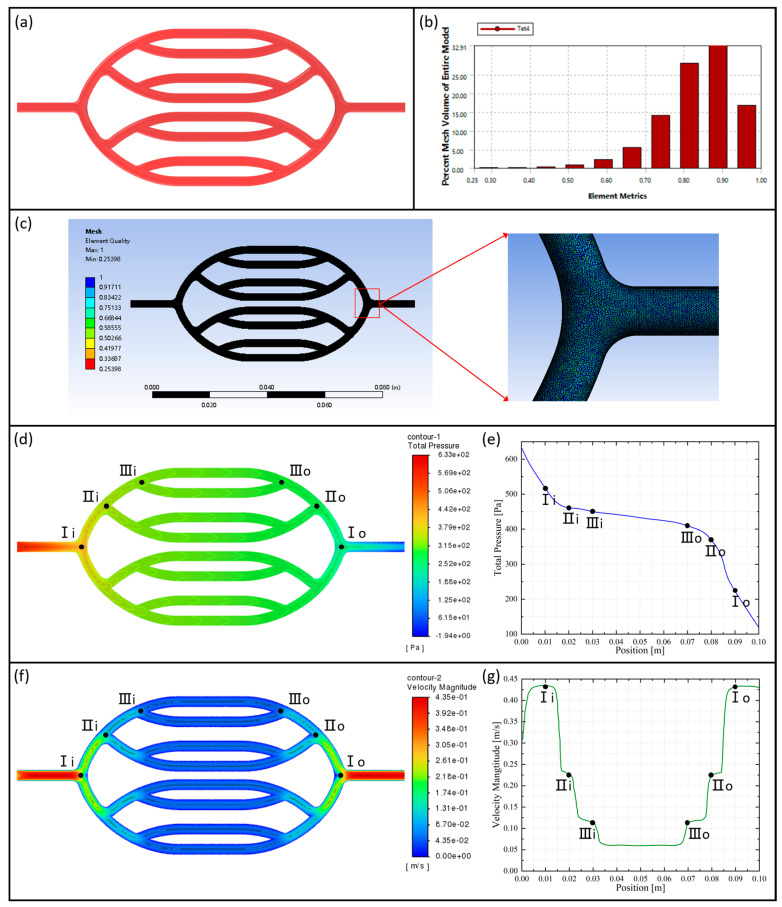
(**a**) Vascular network 3D model, (**b**) element quality statistics, (**c**) overall and local meshing, (**d**) vascular network axial section pressure distribution cloud, (**e**) tributary vascular pressure distribution curve, (**f**) vascular network axial section velocity distribution cloud, and (**g**) tributary vascular velocity distribution curve.

**Figure 6 micromachines-15-00463-f006:**
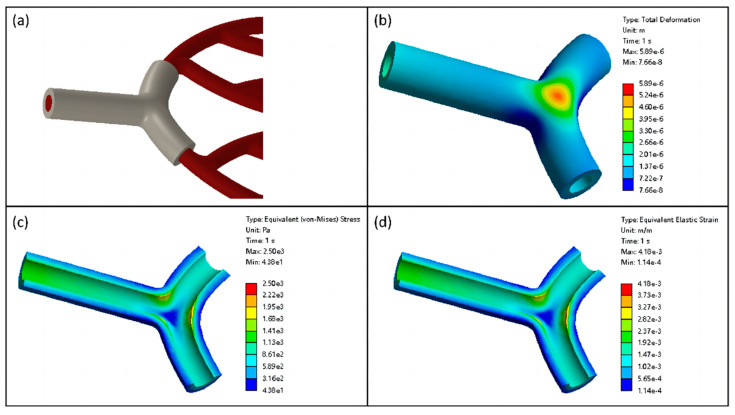
(**a**) Schematic of local coupling, (**b**) blood vessel total deformation cloud, (**c**) blood vessel axial section equivalent stress cloud, and (**d**) blood vessel axial section equivalent elastic strain cloud.

**Figure 7 micromachines-15-00463-f007:**
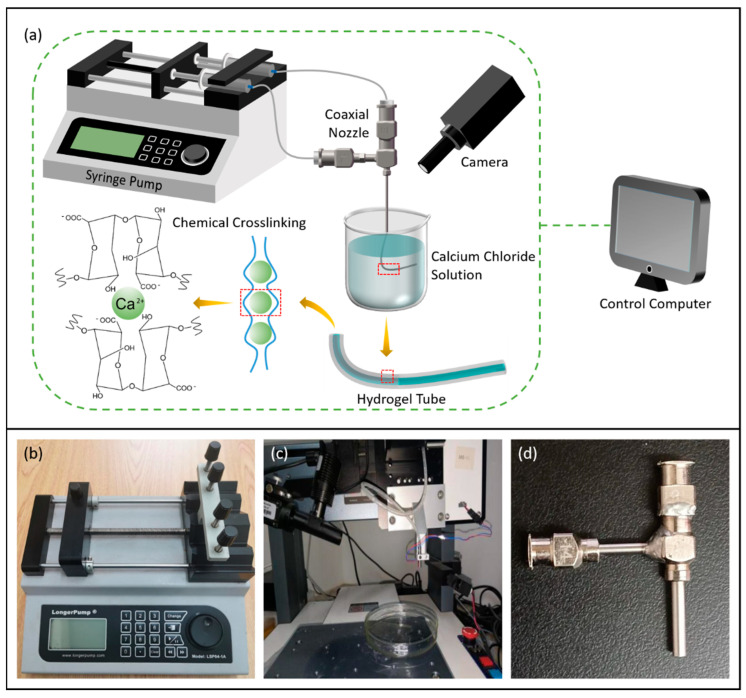
(**a**) Schematic of fabrication process based on coaxial jet, (**b**) peristaltic pump, (**c**) experimental platform, and (**d**) coaxial nozzle.

**Figure 8 micromachines-15-00463-f008:**
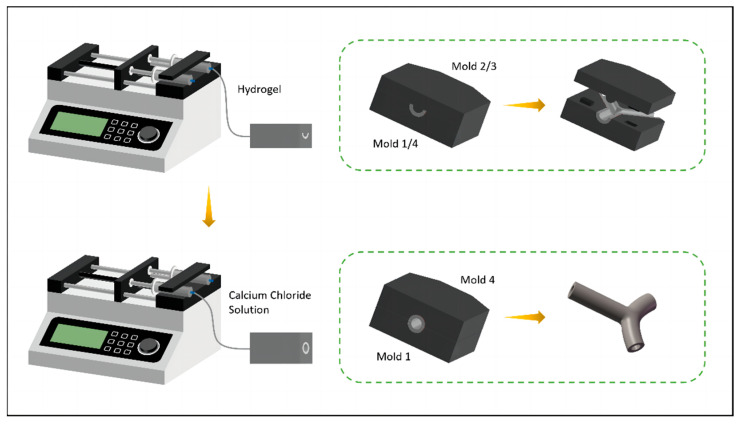
Schematic of fabrication process based on mold method.

**Figure 9 micromachines-15-00463-f009:**
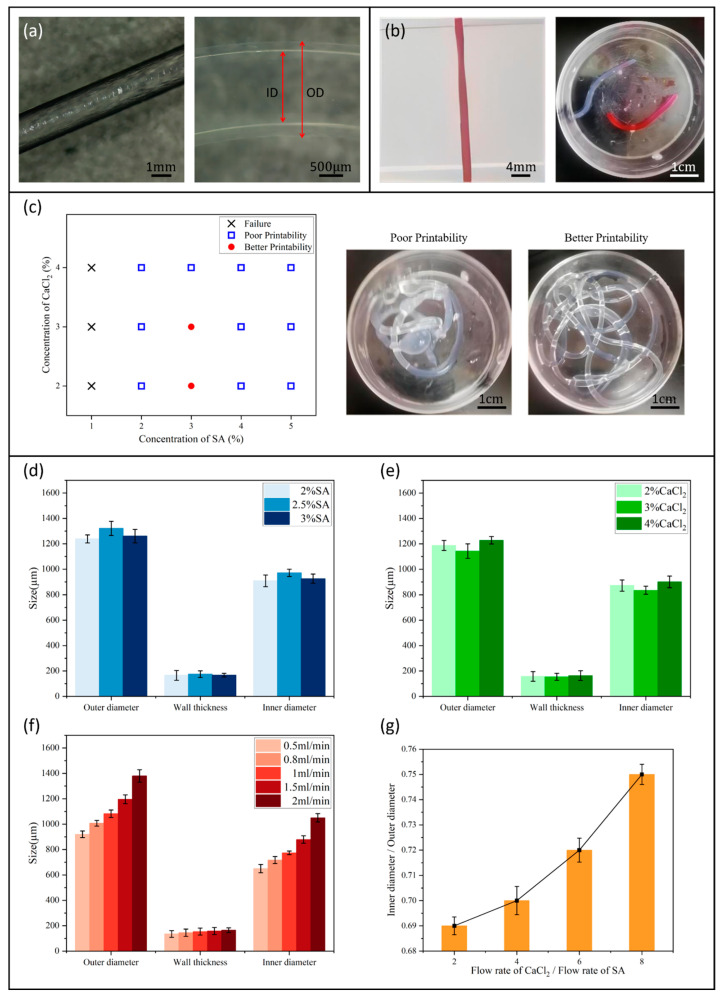
(**a**) Linear extrudate, (**b**) perfusion experiment, (**c**) the printability of vascular scaffolds, (**d**) effect of bioink concentration on duct diameter, (**e**) effect of cross-linking agent concentration on duct diameter, (**f**) effect of the flow rate of bioink on duct diameter, and (**g**) effect of cross-linking agent to bioink flow rate ratio on the ratio of inner to outer duct diameter.

**Figure 10 micromachines-15-00463-f010:**
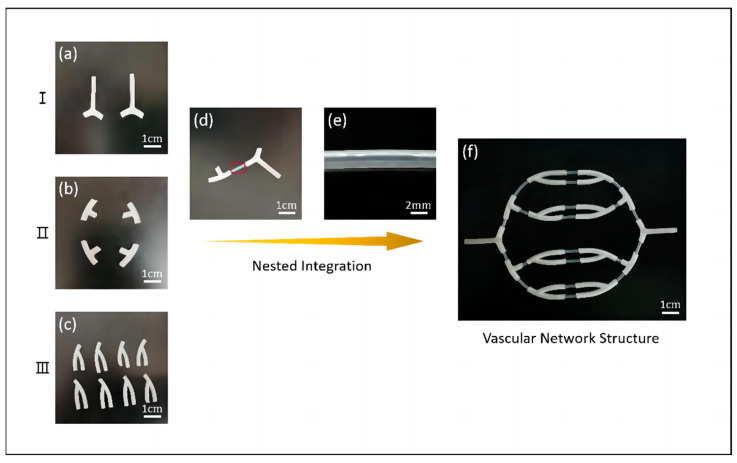
Schematic of the fabrication process of vascular network scaffold. (**a**–**c**) Bifurcation structure at all levels of vascular structures, (**d**,**e**) integration of bifurcated and single-through structures, and (**f**) vascular network structure.

**Figure 11 micromachines-15-00463-f011:**
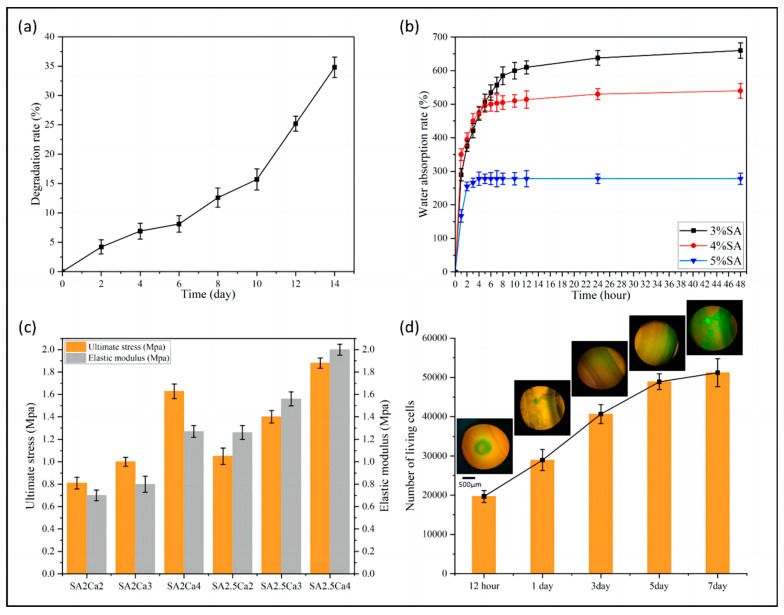
(**a**) Degradation rate curve of vascular scaffolds, (**b**) water absorption rate of vascular scaffolds with different concentrations of bioink, (**c**) ultimate stress and elastic modulus of vascular scaffolds with different concentrations of bioink and cross-linking agents, and (**d**) number of living cells and cell culture observation.

**Table 1 micromachines-15-00463-t001:** Coaxial nozzle dimensions.

Model	Inner Diameter (mm)	Outer Diameter (mm)	Wall Thickness (mm)
14G	1.60	2.10	0.2500
18G	0.90	1.30	0.2000

## Data Availability

The data presented in this study are available upon request from the corresponding author.

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
