# Peer review of "Coaxial 3D Bioprinting Process Research and Performance Tests on Vascular Scaffolds"

_micromachines, 2024, doi:10.3390/mi15040463_

Round 1

Reviewer 1 Report

Comments and Suggestions for Authors

In this work, the authors proposed a method for fabricating vascular tissues based on a coaxial 3D bioprinting technology combined with the mold method. The theoretical models of coaxial jet and vascular network have been studied and the experiments of vascular scaffolds fabrication using the proposed method were carried out. Overall, it was an interesting work with good and clear demonstration. Some major comments were listed as follows:

1.  In this work: https://doi.org/10.1115/1.4063452, an coaxial nozzle assisted embedded 3D printing method was also developed to directly print vascular structures within a matrix material. What's the unique contribution of the authors' method to overcome the limitations of this method? Please discuss in the introduction to emphasize the novelty of the manuscript.

2. Why was it necessary to use mold? Other 3D printing methods, like inkjet printing and material extrusion have made free-standing vascular networks from alginate already. The geometries of mold constrained the design and fabrication flexibility of vascular network. Was this a limitation of this work? How to overcome? Please discuss in detail. 

3. The font size in many figure was too small to read. Please make it larger.

Comments on the Quality of English Language

English in this work was good and easy to read. There were some typos in the manuscript. Please proofread before re-submission.

Reviewer 2 Report

Comments and Suggestions for Authors

This could become an interesting paper, especially due to the realization of the bifurcations. However, important experimental results are missing. The theoretical calculations and simulations described in this manuscript and the experiments performed are somewhat strangely put together. On the one hand, calculations of pressure and flow velocity in a vascular network are shown in Figure 5 and this network is also produced experimentally (Figure 10). However, an attempt to allow fluid to flow through this network is not described; this would be the next logical step that would make the theoretical calculations relevant. The fact that such an experiment is not described in the manuscript suggests that it does not work. Is that the case? Are the leakages too large or do the connections between the bifurcated structures and the single-through structures not hold? The authors should write something about this in the manuscript. They should also describe how these structures are connected. Are they simply plugged together or are they, e.g., glued?
The authors cite results of a "microtensile test" and mention Hooke's law. It sounds as if they have stretched their vessels. This is not clearly described. Why is there no test of pressure resistance? That would be much more interesting.

The authors use the word "biomimetic" without explaining what they mean by it. The vascular network is not really biomimetic, as all vessels have the same diameter and there are no microcapillaries. Therefore, the authors should add their definition of the word "biomimetic" to the introduction.

The production of tubular vascular structures with a coaxial nozzle using alginate and calcium chloride has already been published by others before the authors, in particular by Ibrahim Ozbolat et al. This has not been mentioned by the authors, giving the impression that they were the first to do so. This is not acceptable. This needs to be added in the introduction. Citing Ozbolat et al. in material and methods with the sentence "As it is known from the literature [27], during the extrusion process of manufacturing biomimetic vascular scaffolds, we can see that the rate of droplet deposition is constantly changing." is definitely not sufficient.

The conclusion is actually just a summary, except for the last sentence with the unclear term "biomimetic". A real conclusion is missing and should be added.

The use of the term "cell survival rate" is incorrect. The lowest value in Figure 11d is the survival rate, the increase thereafter is proliferation. Otherwise one would have survival rates of more than 100 % if one waited long enough. Please be more precise.

In the conclusion, the authors state a value of 2 mm/s as the "inlet speed". Is this the black slider speed? If so, please convert it into a value that allows comparisons with the work of other groups. Or is this the speed of the liquid in the "inlet"? What does the value mean?

Figure 9 d + e: Please add the corresponding concentrations for CaCl2 (d) and SA (e).
Figure 9 f:  Please add the corresponding flow rate for CaCl2
Figure 9 g: Is this independent from absolute flow rates? Please add a statement in the results section.

Line 401/402: “The blood flow velocity is approximately symmetrically distributed within the vascular network, with a higher flow velocity at the inlet.”
Is the flow velocity at the inlet higher compared to the outlet? Please be more precise.

Line 406/407: “Because the cross-sectional area of blood vessels remains constant at all levels of the model, the inlet flow velocity remains stable when the inlet flow rate is stable.”  I do not understand this sentence. “the inlet flow velocity remains stable when the inlet flow rate is stable” already sounds like a matter of course to me; why does this depend on the first part of the sentence? Could the authors rephrase this sentence, please?

Line 408-411: “From the velocity distribution curve of the tributary vascular, as shown in Figure 5g, we can learn that as the blood flows into the next level of bifurcation, the cross-sectional area doubles, resulting in a halving of the flow velocity.” This also sounds like a matter of course to me. Do we need Figure 5g to “learn” that the cross-sectional area doubles and the flow velocity is reduced by 50%?

Conclusion: In its current version, I cannot rate this manuscript better than average. Experimental results on perfusion experiments would significantly enhance it.

Reviewer 3 Report

Comments and Suggestions for Authors

How far are you from your goal of implanting this in a person? Could you add some comments about this? Maybe some discussion connecting the theory presented with the experiments more in the conclusions. Did the theory allow you to make a superior device?
